# Universal Frequency Domain Perturbation for Single-Source Domain Generalization

Chuang Liu*
cliu_trans@buaa.edu.cn
Beihang University
Beijing, China

Yichao Cao*
caoyichao@seu.edu.cn
Southeast University
Nanjing, China

Xiu Su†
xiusu@csu.edu.cn
Central South University
Changsha, China

Haogang Zhu†
haogangzhu@buaa.edu.cn
Beihang University
Beijing, China
Hangzhou International
Innovation Institute,
Beihang University
Hangzhou, China

## Abstract

In this work, we introduce a novel approach to single-source domain generalization (SDG) in medical imaging, focusing on overcoming the challenge of style variation in out-of-distribution (OOD) domains without requiring domain labels or additional generative models. We propose a **Uni**versal **Freq**uency Perturbation framework for **SDG** termed as *UniFreqSDG*, that performs hierarchical feature-level frequency domain perturbations, facilitating the model's ability to handle diverse OOD styles. Specifically, we design a learnable spectral perturbation module that adaptively learns the frequency distribution range of samples, allowing for precise low-frequency (LF) perturbation. This adaptive approach not only generates stylistically diverse samples but also preserves domain-invariant anatomical features without the need for manual hyperparameter tuning. Then, the frequency features before and after perturbation are decoupled and recombined through the Content Preservation Reconstruction operation, effectively preventing the loss of discriminative content information. Furthermore, we introduce the Active Domain-variance Inducement Loss to encourage effective perturbation in the frequency domain while ensuring the sufficient decoupling of domain-invariant and domain-style features. Extensive experiments demonstrate that *UniFreqSDG* increases the dice score by an average of 7.47% (from 77.98% to 85.45%) on the fundus dataset and 4.99% (from 71.42% to 76.73%) on the prostate dataset compared to the state-of-the-art approaches.

## CCS Concepts

• **Computing methodologies** → **Computer vision**.

---

*Both authors contributed equally to this research.
†Corresponding author.

---

## Keywords

single domain generalization; frequency domain learning; medical image segmentation

**ACM Reference Format:**
Chuang Liu, Yichao Cao, Xiu Su, and Haogang Zhu. 2024. Universal Frequency Domain Perturbation for Single-Source Domain Generalization. In *Proceedings of the 32nd ACM International Conference on Multimedia (MM '24), October 28-November 1, 2024, Melbourne, VIC, Australia.* ACM, New York, NY, USA, 10 pages. https://doi.org/10.1145/3664647.3681536

## 1 Introduction

In deep learning, *Domain Adaptation* (DA) [21, 31] and *Domain Generalization* (DG) [29, 75] aim to address data distribution variations that deviate from the independent and identically distributed (*i.i.d.*) assumption [7]. The objective of DG [15] is to develop a model using data from one or several related but distinct source domains, ensuring robust generalization to any out-of-distribution (OOD) target domain. *Single-source Domain Generalization* (SDG) focuses on the realistic challenge of developing methods that generalize from a single source to multiple OOD target domains[35, 62]. This is particularly significant in medical imaging, where the broad applicability of models in real-world settings is crucial, especially under conditions of limited data availability or privacy concerns[45].

In SDG, *Data Augmentation* is recognized as a key technique, involving the transformation of original $(x, y)$ pairs into $(A(x), y)$ pairs where $A(\cdot)$ simulates domain shifts[48, 58]. The strategic development of $A(\cdot)$ is critical for enhancing model performance. Data augmentation aims to increase the diversity and informativeness of training data, thereby strengthening the model's ability to generalize across OOD target domains.

Data augmentation techniques for SDG can broadly be categorized into three types: *Image Transformations*, *Model-based Augmentation*, and *Feature-based Augmentation* [62, 75]. *Image Transformations* are widely used but often suffer from being problem-specific and limited in simulating domain shifts [51, 70]. Model-based Augmentation employs either off-the-shelf or learnable models $A(\cdot)$ for adversarial learning, aiming to maximize differences between source and synthesized samples while ensuring semantic consistency [48, 58, 64, 74]. However, challenges arise in dense prediction tasks and in maintaining semantic details in high-quality synthetic medical images[48]. *Feature-based Augmentation* attempts to expand the single source domain by generating pseudo domains [32, 47, 62, 72]. However, to effectively balance domain diversity

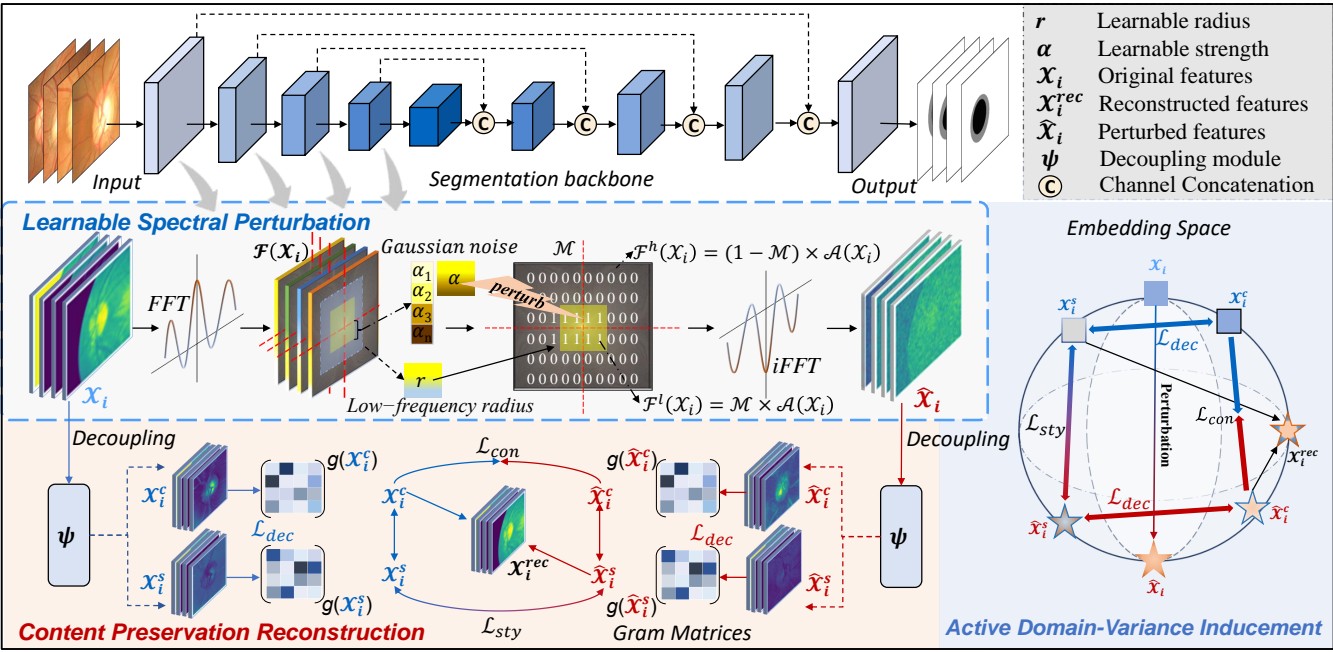

**Figure 1: The overall framework of the proposed UniFreqSDG. Our proposed framework comprises three key components: (a) LSP introduces learnable LF radius $r$ and Gaussian perturbation strength $\alpha$ to adaptively enlarge the scope of single-source domain features. (b) CPR decouples and recombines pre- and post-perturbation features to preserve the content information and asisit the model to perceive the potential domain shifts. (c) The ADI designs three loss function $\mathcal{L}_{dec}$, $\mathcal{L}_{sty}$ and $\mathcal{L}_{con}$ to ensure the effectiveness of perturbation strength and information decoupling.**

and maintain semantic integrity, domain expansion primarily requires careful calibration of complex loss term interactions across input, latent, and output spaces. Some other approaches [13, 33, 77] include using mix-up or randomization strategies on latent feature statistics to enhance data diversity, but these rely on fixed perturbations (linear interpolation or random perturbation), limiting effective domain transfer.

Recently, frequency domain methods oriented towards multisource domain generalization in the tasks of classification [53, 61, 63, 67] and semantic segmentation [28, 36, 54, 79] have elucidated several intriguing phenomena: *i)* Among the amplitude and phase spectra, the phase spectrum is more adept at capturing high-level semantics. *ii)* The low-frequency (LF) bands of the amplitude spectrum predominantly harbor style information or low-level statistics, such as illumination and lighting. Nevertheless, these methods face significant challenges: *i)* They necessitate more stringent requirements on the quantity and labeling of source domains, thus limiting the widespread deployment in practical settings. *ii)* The simple linear combination of known style components results in samples that are significantly inadequate in covering the extensive OOD target domains.

To ameliorate the limitations regarding universality, flexibility, and generalization encountered by prior methodologies, as shown in Fig. 1, we propose a Universal Frequency Perturbation framework for SDG termed as ***UniFreqSDG***. The core idea lies in the infusion of adaptive frequency domain perturbations into the multi-level

latent space of training data derived from a singular source domain. The contributions of this work can be summarized as follows:

- We propose a universal adaptive frequency domain perturbation approach for single-source domain generalization in medical imaging. Our method enhances adaptability and robustness by applying hierarchical feature-level frequency perturbations directly, without needing domain labels or additional generative models.
- We introduce the Learnable Spectral Perturbation (LSP) module, which adaptively learns sample frequency distributions to predict low-frequency (LF) perturbation range and strength automatically. This method can generate stylistically diverse samples while preserving domain-invariant features.
- We design the Content Preservation Reconstruction (CPR) strategy to prevent the loss of content information. Additionally, the Active Domain-variance Inducement (ADI) Loss is introduced to ensure the effectiveness of each feature component and the "Decoupling-Reconstruction" process.
- Extensive experiments on UniFreqSDG achieve a remarkable 7.47% increase in dice score for the fundus dataset, reaching 85.45%, and a 4.99% improvement on the prostate dataset, achieving 76.73%. These results not only signify substantial progress but also surpass previous state-of-the-art methods in SDG task.

## 2 Related Work

### 2.1 Domain Generalization in Medical Imaging

In the medical image segmentation, domain generalization (DG) [1, 22, 56, 69] has been explored extensively. Some techniques[36, 37, 51, 70, 80] employ image-level data augmentation for style diversification, while others [8, 13, 42, 79] manipulate feature space samples through methods like adversarial learning or statistics randomization. Meanwhile, strategies such as [20, 27] focusing on learning cross-domain feature invariance through a distanglement manner. Recently, frequency-based methods [36, 57, 79] have begun to emerge in the field of medical imaging. However, the significant style variations between different imaging modalities limit the universality of these methods. This motivated us to explore an effective and general frequency perturbation approach in the feature latent space utlizing an adaptive frequency perturbation method.

### 2.2 Single-source Domain Generalization (SDG)

SDG method [18, 35, 40] focuses on extracting robust, invariant features from only the source data, assuming no access to the target domain. This poses challenges due to limited data diversity. Standard methods include Image Transformations, Adversarial Learning, Model-based Augmentation, and Feature-based Augmentation. Notable developments include diverse data augmentation techniques [29, 66, 77] to reduce overfitting from domain shifts. Adversarial approaches [9, 48, 73, 74] use an adversarial domain synthesizer (ADS) for creating new domains via interpolation and ensuring semantic consistency through mutual information regularization. In medical imaging, techniques like image-level augmentation [51] and adversarial training [8, 65] are used to simulate unseen images and extend target domain coverage [11, 12, 30, 39, 43, 60, 68]. These strategies guide our pursuit of a more universal and adaptable SDG method by focusing on feature-level data augmentation to enhance model resilience to diverse domain features.

### 2.3 Data Augmentation

Data augmentation, a key strategy for SDG, includes Image Transformations, Adversarial Learning, Model-based Augmentation, and Feature-based Augmentation [75]. Traditional image transformations [51, 70] often demand elaborate manual design, whereas Model-based Augmentation [64, 74, 76] typically employs Style Transfer Models or Learnable Augmentation Networks. Techniques like AdaIN facilitate image transfer across domains for augmentation. Feature-based Augmentation, in contrast, avoids complex image-to-image models, focusing instead on efficient architecture. Feature-based data augmentation methods [13, 46, 77, 78] strategically normalize or randomize the statistics of features across domains to improve adaptability to domain shifts. Frequency-based perturbations [10, 63] and Amplitude Mixup (AM) in medical tasks [36, 79] innovate by blending LF components from different sources, though these often face challenges due to the limited diversity of linearly mixed styles and multiple source requirements [2–6, 38]. Our novel LSP module circumvents these limitations by automatically learning perturbation regions and strengths in feature spectra, providing a plug-and-play solution for enhancing base models.

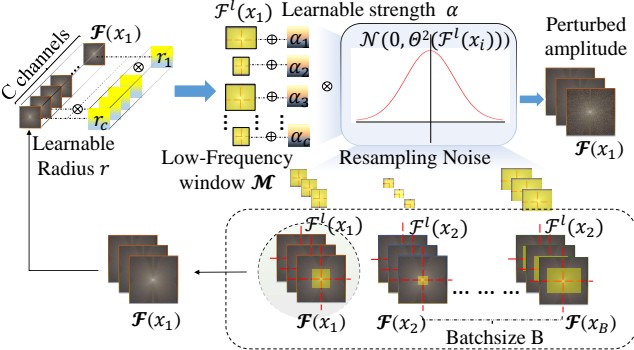

**Figure 2: Each feature map learns a dynamic LF radius $r$ and strength $\alpha$ to obtain the LF perturbations. Then, the dynamically changing Gaussian noise is injected into the learned LF region.**

## 3 Methodology

### 3.1 Problem Definition and Method Overview

In single-source domain generalization, the objective is to train a model, symbolized as $\theta : \mathcal{X} \rightarrow \mathcal{Y}$, using a sole source domain $\mathcal{D}_s = \{x_i^s, y_i^s\}_{i=1}^{N_s}$ (with $N_s = 1$ in SDG). The expectation is that this model will perform effectively across multiple target domains $\mathcal{D}_t = \{x_i^t\}_{i=1}^{N_t}$, where $N_t > 1$. The primary challenge in SDG lies in the unpredictable nature of these target domains. These domains often exhibit substantial variations from the training domain, especially in aspects like style, contrast, and overall visual properties.

### 3.2 Learnable Spectral Perturbation (LSP)

This section introduces a flexible frequency domain perturbation strategy, termed *Learnable Spectral Perturbation* (LSP), which generates stylistically diverse counterparts from single-domain features. As shown in Fig. 2, LSP adapts to the input data by learning the range of LF representations and dynamically adjusting this range based on frequency domain data, thus enhancing the diversity of representation styles from a single source domain. The forthcoming subsections will explore this approach in detail, focusing on the learnable LF window and the perturbation strength.

*3.2.1 Learnable Low-frequency window.* For a given intermediate feature $\mathcal{X}_i \in \mathbb{R}^{H \times W \times C}$, with $H$, $W$, and $C$ denoting the height, width, and number of channels respectively, we perform a 2D FFT [14] for each channel independently to obtain the corresponding frequency representations $\mathcal{F}(\mathcal{X}_i) \in \mathbb{R}^{H' \times W' \times C'}$. This computational process can be formalized as follows:

$$\mathcal{F}(\mathcal{X}_i)(u,v,c) = \sum_{h=0}^{H-1} \sum_{w=0}^{W-1} \mathcal{X}_i(h,w,c) e^{-j2\pi \left( \frac{h}{H} u + \frac{w}{W} v \right)} \quad (1)$$

where $j^2 = -1$. Here, $u$, $v$, and $c$ respectively denote the indices in the frequency domain representation $\mathcal{F}(\mathcal{X}_i)$ along the height, width, and channel dimensions. Meanwhile, the amplitude and phase components are then respectively expressed as:

$$\mathcal{A}(\mathcal{X}_i)(u,v,c) = \left[ R^2(\mathcal{F}(\mathcal{X}_i)(u,v,c)) + I^2(\mathcal{F}(\mathcal{X}_i)(u,v,c)) \right]^{1/2}$$

$$\mathcal{P}(\mathcal{X}_i)(u,v,c) = \arctan \left[ \frac{I(\mathcal{F}(\mathcal{X}_i)(u,v,c))}{R(\mathcal{F}(\mathcal{X}_i)(u,v,c))} \right] \quad (2)$$

where $R(\cdot)$ and $I(\cdot)$ represent the real and imaginary part of $\mathcal{F}(X_i)$, respectively. For intermidiate feature maps $X_i \in \mathbb{R}^{H \times W \times C}$, the Fourier transformation for each channel is computed independently to get the corresponding amplitude and phase information. Besides, thanks to the conjugate symmetric property of FFT, $\mathcal{F}(X_i)$ only needs retain the half of spatial dimensions thus has a spatial resolution of $H \times \left( \left\lfloor \frac{W}{2} \right\rfloor + 1 \right) \times C$.

In order to implement perturbations within the LF range, we first employ a learnable binary window $\mathcal{M}$ to separate LF and HF components. The window function is defined as:

$$\mathcal{M}(u,v,c) = \begin{cases} 1, if \; |u - W'/2| \leq rW' \text{ and } |v - H'/2| \leq rH' \\ 0, \quad \text{otherwise} \end{cases} \tag{3}$$

where $r$ denotes the proportion of the LF range within the frequency domain representation $\mathcal{F}(X_i)$. The role of $\mathcal{M}$ is to introduce a learnable binary mask near the center of $\mathcal{F}(X_i)$, which adaptively adjusts its range based on the input data. This mask defines the LF range crucial for frequency domain perturbation. Notably, the LF range $r$ is independently learned and computed for different channels of the input features $X_i$ at various levels.

To calculate the LF range $r$, we begin by aggregating the concatenated amplitude composition $concat[\mathcal{A}(X_i), \mathcal{A}(X_i)]$ via a global pooling layer, resulting in a semantically-rich channel vector $\{\alpha\}_{i=1}^{2c} = [\alpha_1, \alpha_2, \ldots, \alpha_{2c}]$. An *MLP* consisting of fully connected layers and activation functions then maps this vector to a channel-wise feature space $\mathcal{S} = [\beta_1, \beta_2, \ldots, \beta_{2c}]$, indicative of the learnable perturbation ratio $r$ and strength $\alpha$. The computational process is as follows:

$$\mathcal{S} = Expand \left( MLP \left( Pooling \left( Concat[\mathcal{A}(X_i), \mathcal{A}(X_i)] \right) \right) \right) \tag{4}$$

The *Expand* function scales vectors to match the spatial dimensions of position embedding features. Eq. (4) allows us to observe changes in the input sample's spectrum, $\mathcal{F}(X_i)$, and to introduce adaptive Gaussian perturbations with strength $\alpha = \mathcal{S}[c : 2c]$ into the LF components across varying radius $r = \mathcal{S}[0 : c]$. This approach ensures that the model has considerable flexibility in handling the input features.

With the learnable LF mask $\mathcal{M}$ applied to each feature channel, we can delineate the low-frequency (LF) and high-frequency (HF) components in the frequency domain as follows:

$$\mathcal{F}^l(X_i) = \mathcal{M} \odot \mathcal{A}(X_i) \tag{5}$$

$$\mathcal{F}^h(X_i) = (1 - \mathcal{M}) \odot \mathcal{A}(X_i) \tag{6}$$

where $\odot$ denotes element-wise multiplication. In Fig. 3, we visualize the learned low-frequency masks $\mathcal{M}$ as well as the feature and spectral representations pre- and post-perturbation.

*3.2.2 Learnable Perturbation Strength.* For a feature map $X_i$, we separate its frequency domain into low- and high-frequency components, $\mathcal{F}^l(X_i)$ and $\mathcal{F}^h(X_i)$, as defined in Eq. (5) and (6). Recognizing the varied intensities of elements due to domain shifts in different data distributions, we aim to inject adaptive noise into the LF components that represent style features. This is preceded by a statistical analysis of the frequency distribution of the data, where the LF spectrum is modeled as a multivariate Gaussian distribution, centered at the original values with variance derived from the values across different samples.

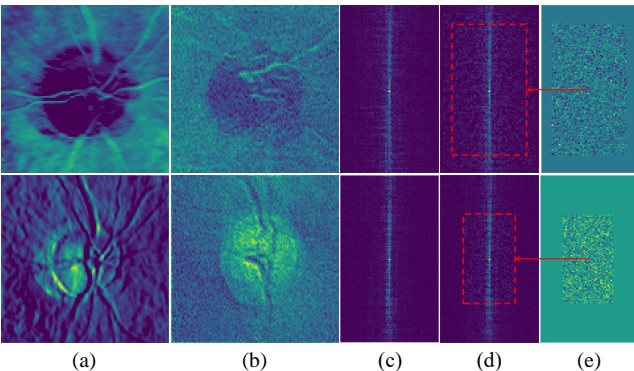

|     |     |     |     |     |
| (a) | (b) | (c) | (d) | (e) |

**Figure 3: We visualized the perturbation of the encoder's first-block of *UniFreqSDG$_m$*: (a) Original input features, (b) features after perturbation, (c) spectrum of original features, (d) spectrum after perturbation, (e) learned LF regions and noise.**

$$\Theta^2(\mathcal{F}^l(X_i)) = \frac{1}{W'H'r^2} \sum_u \sum_v \left[ \mathcal{F}^l(X_i) - \mathbb{E}[\mathcal{F}^l(X_i)] \right]^2 \tag{7}$$

where the $W'H'r^2$ denotes the area of the LF region. The variance magnitude $\Theta^2$ reflects the intensity of element variation, serving as a crucial metric for understanding domain shifts. This quantitative analysis delineates the fluctuation of elements in response to domain changes. A higher variance $\Theta^2$ indicates increased variability, underscoring element sensitivity to domain-specific modifications. Noise perturbations are then injected into the LF components $\mathcal{F}^l(X_i)$, based on the variation intensity $\Theta^2$ across different frequency ranges. This process enhances adaptability to the input data, as described below:

$$\widehat{\mathcal{F}}^l(X_i) = \mathcal{F}^l(X_i) + \alpha \cdot \zeta, \quad \zeta \sim \mathcal{N}(0, \Theta^2(\mathcal{F}^l(X_i))) \tag{8}$$

where the parameter $\zeta$ denotes the Gaussian noise sampled from the normal distribution, and $\alpha \in \mathbb{R}^{B \times C}$ is a matrix of channel scores that determine the perturbation strength calculated in Eq. (4).

Finally, through the inverse Fast Fourier Transform (*iFFT*), we combine the phase and the spectrum of the perturbed LF components $\widehat{\mathcal{F}}^l(X_i)$ with those of the LF components $\mathcal{F}^h(X_i)$ to obtain the final perturbed features:

$$\widehat{X}_i = \mathcal{F}^{-1}([\widehat{\mathcal{F}}^l(X_i), \mathcal{F}^h(X_i)]) \tag{9}$$

Thus, we successfully expand the single-source domain training data with an effectively, learnable spectral perturbation module.

### 3.3 Content Preservation Reconstruction (CPR)

To prevent damage to the content components during high-intensity feature perturbations, we developed a Content Preservation Reconstruction strategy. This approach involves decomposing both the pre-perturbation and post-perturbation feature maps, $X_i$ and $\widehat{X}_i$, into content and style components. The style features from post-perturbation are then recombined with the content features from pre-perturbation to train the model. The benefits of this approach are twofold: on the one hand, it replenishes any potential loss in the content part of the model; on the other hand, it guides the model to understand the perturbation information. This strategy can be summarized as "Perturbation Introduction & Content Preservation".

---

**Algorithm 1** Framework for UniFreqSDG Method.

---

1: **Input:** Pre-trained backbone, SDG dataset, maximum training epochs $N$, LSP module shown in Fig. 2.
2: Init $\tau = 0$, parameters of LSP;
3: **while** $\tau \leq N$ **do**
4:     (1) Hierarchical feature map extraction for input: $X_i$;
5:     (2) $\widehat{X}_i \leftarrow LSP(\mathcal{F}(X_i))$ ;                  ▷ Perturbation Injection
6:     (3) Generate the content-related weight tensor $\mathcal{W}_i$;
7:     $X_i^{rec} = (\mathcal{W}_i \odot X_i) + (X_i - X_i^c)$ ;                  ▷ CPR Strategy
8:     (4) ADI strategy for diverse feature component:
9:     $\mathcal{L}_{adi} = \lambda_1 \mathcal{L}_{dec} + \lambda_2 \mathcal{L}_{sty} + \lambda_3 \mathcal{L}_{con}$ ▷ ADI Loss Calculation
10:    (5) Calculate the overall loss:
11:    $\mathcal{L}_{total} = \lambda_{seg} \mathcal{L}_{seg} + \lambda_{adi} \mathcal{L}_{adi}$ ;
12:    (6) Optimize the learnable weights of UniFreqSDG;
13: **end while**
14: **Output:** The optimized weights of UniFreqSDG.

---

Specifically, we decouple the original frequency domain features $X_i$ into category-related content components $X_i^c$ and domain-related style components $X_i^s$:

$$X_i^c = \mathcal{W}_i \odot X_i, X_i^s = X_i - X_i^c \tag{10}$$

where $\mathcal{W}_i$ represents a weight tensor learned by the decoupling module. In this work, we provide an example of a simple decoupling module, which consists of channel attention ($CA$) and spatial attention ($SA$) [59], with the aim of filtering generalizable content information from both the channel and spatial perspectives. The process can be simplified to $\mathcal{W}_i = CA(X_i) \odot SA(X_i)$. Additionally, we conduct an ablation study to assess various attention types for the decoupling operation in the experiment section. While the architecture design of the attention module is not the primary focus of our current work, it presents a valuable avenue for future exploration. Consequently, specific information within $X_i$ can be directly extracted through element-wise multiplication.

Similarly, the perturbed features $\widehat{X}_i$ can be decoupled into content components $\widehat{X}_i^c$ and domain-related style components $\widehat{X}_i^s$:

$$\widehat{X}_i^c = \widehat{\mathcal{W}}_i \odot \widehat{X}_i, \widehat{X}_i^s = \widehat{X}_i - \widehat{X}_i^c \tag{11}$$

where $\widehat{\mathcal{W}}_i$ denotes the weight tensor corresponding to the perturbed features $\widehat{X}_i^c$.

To compensate for the potential loss of content components due to perturbations in the frequency domain, we design a feature reconstruction strategy focused on content preservation. This strategy involves recombining the perturbed style components with the original content components:

$$X_i^{rec} = X_i^c + \widehat{X}_i^s \tag{12}$$

In this way, the model's decoder is able to obtain well-represented content alongside hierarchical features $X^{rec}$ perturbed with a certain strength of style disturbance. This capability facilitates the model's adaptation to varying degrees of domain shift problems throughout the training process.

## 3.4  Active Domain-variance Inducement (ADI)

During the model's optimization, we implemented a strategy named *Active Domain-variance Inducement* to ensure the convergence of

intricately designed structures. This optimization strategy is based on three key objectives: *i)* to decouple the features into content and style components effectively during the reconstruction process; *ii)* to maximize the difference in style components post-perturbation compared to pre-perturbation; *iii)* to maintain as much consistency as possible in the content components post-perturbation with those pre-perturbation.

*3.4.1  Gram-based Style Discrepancy Metric.* To maximize the domain discrepancy between style components before and after perturbation, we introduce a Gram-matrix based metric to explicitly depict the domain discrepancy. The Gram matrix captures stylistic features like textures and patterns, indirectly delineating domain-specific information without direct relevance to task objectives [19]. First, to promote the full decoupling of features into content and style components before and after perturbation, our decouple process can be presented as follow:

$$\mathcal{L}_{dec} = -\frac{1}{B} \sum_{i=1}^{B} \left( \left( \mathcal{G}(X_i^s) - \mathcal{G}(X_i^c) \right)^2 + \left( \mathcal{G}(\widehat{X}_i^s) - \mathcal{G}(\widehat{X}_i^c) \right)^2 \right) \tag{13}$$

where $\mathcal{G}(X_i)$ denotes the Gram matrix of the $i$-th samples in $X$ respectively. The Gram matrix $\mathcal{G}(X)$ is calculated as:

$$\mathcal{G}(X) = \frac{1}{C \times N} X X^T \tag{14}$$

$X \in \mathbb{R}^{H \times W \times C}$ is reshaped into a matrix of dimensions $(C, N)$, with $C$ being the number of channels and $N$ the total number of elements divided by the number of channels, and $X^T$ is the transpose of $X$. Secondly, to encourage a significant difference between the style components after perturbation compared to those before, we set a perturbation inducement term:

$$\mathcal{L}_{sty} = -\frac{1}{B} \sum_{i=1}^{B} \left( \mathcal{G}(X_i^s) - \mathcal{G}(\widehat{X}_i^s) \right)^2 \tag{15}$$

*3.4.2  Explicit Content Consistency Constraint.* Additionally, the perturbation injection process involves learnable LF ranges and perturbation strength. To facilitate representation, we define a feature similarity measurement function, *Sim* (e.g., Cosine Similarity). The following constraint rules are based on this measurement. These learnable perturbation noises should consider the preservation of content components:

$$\mathcal{L}_{con} = Sim(X_i^c, \widehat{X}_i^c) \tag{16}$$

The overall Active Domain-variance Inducement loss, $\mathcal{L}_{adi}$, can be defined as:

$$\mathcal{L}_{adi} = \lambda_1 \mathcal{L}_{dec} + \lambda_2 \mathcal{L}_{sty} + \lambda_3 \mathcal{L}_{con} \tag{17}$$

## 3.5  Training Objective

Our proposed **UniFreqSDG** method is summarized in Algorithm 1. In our methodology, three distinct loss functions are employed: segmentation loss $\mathcal{L}_{seg}$ and Active Domain-variance Inducement loss $\mathcal{L}_{adi}$. The segmentation loss $\mathcal{L}_{seg}$ comprises a combination of Dice loss $\mathcal{L}_{dice}$ and Cross-Entropy loss $\mathcal{L}_{ce}$. Thus, our overall optimization objective can be expressed by the following equation:

$$\mathcal{L}_{seg} = \mathcal{L}_{dice} + \mathcal{L}_{ce}, \mathcal{L}_{total} = \lambda_{seg} \mathcal{L}_{seg} + \lambda_{adi} \mathcal{L}_{adi} \tag{18}$$

**Table 1: DSC Comparison of State-Of-The-Art (SOTA) methods on Fundus segmentation task [13]. We mark the top results in bold.**

| Methods | Optical Disc / Cup Segmentation (DSC↑) | | | | | Avg. DSC ↑ |
|---|---|---|---|---|---|---|
| | A to Rest | B to Rest | C to Rest | D to Rest | E to Rest | |
| ERM [50] | 74.54, 59.21 | 82.30, 71.96 | 78.06, 59.12 | 79.79, 59.23 | 85.25, 58.88 | 70.83 |
| RandConv [66] (ICLR'21) | 79.63, 64.14 | 85.00, 72.40 | 87.77, 69.57 | 83.08, 64.38 | 86.31, 60.37 | 75.27 |
| CSDG [42] (TMI'22) | 78.40, 65.11 | 86.02, 76.19 | 87.64, 70.79 | 83.51, 65.26 | 87.09, 65.28 | 76.53 |
| MaxStyle [8] (MICCAI'22) | 77.40,65.44 | 86.95,74.52 | 87.95, 67.62 | 84.69, 66.05 | 87.95, 64.84 | 76.34 |
| SAN-SAW [44] (CVPR'22) | 76.42, 59.01 | 83.79, 73.23 | 84.17, 65.51 | 81.83, 62.36 | 87.00, 64.42 | 73.77 |
| EFDM [71] (CVPR'22) | 78.79,57.73 | 84.83,72.20 | 85.25, 65.94 | 82.13, 61.62 | 85.45, 63.02 | 73.70 |
| DSU [33] (ICLR'22) | 76.88, 61.26 | 84.17, 74.10 | 89.12, 70.16 | 83.53, 63.19 | 87.09, 59.65 | 74.91 |
| SLAug [51] (AAAI'23) | 79.83, 64.53 | 87.42, 75.94 | 88.18, 71.30 | 83.17, 64.52 | 86.57, 67.12 | 76.86 |
| TriD [13] (MICCAI'23) | 81.86, 66.67 | 88.19, 75.43 | 89.62, 70.85 | 84.81, 67.53 | 87.88, 66.96 | 77.98 |
| Aloft-E [23] (CVPR'23) | 87.46, 74.67 | 89.25, 76.53 | 90.32, 74.85 | 88.45, 75.53 | 88.31, 76.63 | 82.20 |
| $UniFreqSDG_s$ | $(\mathbf{85.83, 71.38})_{\pm0.54}$ | $(\mathbf{89.16, 75.67})_{\pm0.21}$ | $(86.59 , 74.95)_{\pm0.47}$ | $(\mathbf{87.12, 74.32})_{\pm0.98}$ | $(87.59 , 75.17)_{\pm0.36}$ | **80.78** |
| | +3.97, +4.71 | +0.97, +0.24 | -3.03, +4.10 | +2.31, +6.79 | -0.29, +8.21 | +2.8 |
| $UniFreqSDG_m$ | $(\mathbf{90.03, 77.61})_{\pm0.65}$ | $(\mathbf{92.89, 79.65})_{\pm1.18}$ | $(91.20 , 78.48)_{\pm0.32}$ | $(\mathbf{91.76, 78.96})_{\pm1.23}$ | $(90.98 , 79.64)_{\pm0.56}$ | **85.31** |
| | +8.17, +10.94 | +4.70, +4.22 | +1.58, +7.63 | +6.95, +11.43 | +3.10, +12.68 | +7.33 |
| $UniFreqSDG_l$ | $(\mathbf{91.05, 76.59})_{\pm0.34}$ | $(\mathbf{92.59, 81.91})_{\pm0.92}$ | $(91.59 , 78.86)_{\pm0.26}$ | $(\mathbf{91.87, 79.39})_{\pm0.78}$ | $(91.08 , 80.24)_{\pm0.43}$ | **85.45** |
| | +9.19, +9.92 | +4.40, +6.48 | +1.97, +8.01 | +7.06, +11.86 | +3.20, +13.28 | +7.80 |

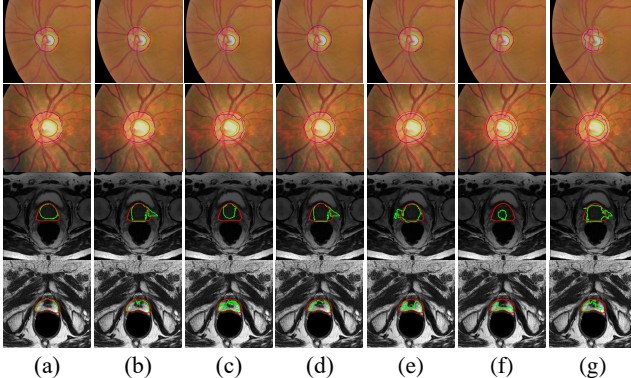

(a) (b) (c) (d) (e) (f) (g)

**Figure 4: Qualitative comparisons across DG Methods: fundus imaging (top rows) vs. Prostate (bottom rows) with ground truth (red contours) and Predictions (green contours). The subfigures (a) to (g) correspond to: (a) our UniFreqSDG, (b) Aloft-E [23], (c) TriD [13], (d) SLAug [51], (e) CSDG [42], (f) MaxStyle [8], and (g) RandConv [66].**

where $\lambda_{seg}$ and $\lambda_{adi}$ are hyperparameters to balance the weights of the segmentation loss and the inducement loss, respectively. A detailed discussion of the hyperparameters for the loss function is provided in ablation studies section. During inference, all the perturbation operations are removed, and the segmentation network is tested on the unseen target domains.

## 4 Experiments

### 4.1 Datasets and Evaluation Metrics

We performed Single-source Domain Generalization (SDG) experiments across three datasets: OD/OC (Fundus) segmentation dataset [13], Prostate segmentation dataset [56], and the PACS dataset [48] for natural image classification. We utilized Dice Similarity Coefficients (DSC) [%] and Average Surface Distance (ASD) [pixel] to quantitatively assess the segmentation results. Detailed descriptions of each dataset and the ASD results are presented in the Appendix. ResNet [24] is utilized as the backbone for the segmentation

network, modified with UNet-style skip connections. The evaluation methods for the model are consistent with those established in previous research [49, 58]. In the ablation studies, we utilize $UniFreqSDG_m$ to perform all the experiments. More details about experimental hyperparameters are shown in Appendix.

### 4.2 Main Results

We conducted a comprehensive evaluation of our proposed method in comparison with state-of-the-art SDG methods, such as Aloft-E [23], TriD [13], SLAug [51] and MaxStyle [8], on medical segmentation datasets. We designed three variants of different scales, tailored to different computational capacities, to ensure comprehensive experimentation: $UniFreqSDG_s$, $UniFreqSDG_m$, and $UniFreqSDG_l$, based respectively on ResNet-18, ResNet-34, and ResNet-50. Previously, most existing methods utilize the ResNet-34 architecture. The results of this comparison are presented in Tab. 1 and Tab. 2.

For instance, from Tab. 1, we observe that $UniFreqSDG_m$ achieves a substantial increase in the Dice Similarity Coefficient (DSC) for OD/OC segmentation tasks in Domain A generalization to other domains, with improvements of 8.17 and 10.94, respectively, compared to the TriD method [13]. Furthermore, similar results are observed in other settings within Tab. 1 and across Tab. 2. These experimental results validate the effectiveness of our proposed Universal Frequency Domain Perturbation method for SDG tasks. To further validate the model's generalizability to domain shift problems, we conducted SDG classification tasks on the Natural image classification dataset PACS [29, 48]. UniFreqSDG is trained on one of the four style source domains—Photo (P), Art (A), Cartoon (C), and Sketch (S)—and then tested on the remaining domains. The related results can be seen from Tab. 3, our approach achieves a significant advantage on three different backbones.

### 4.3 Ablation Studies

*4.3.1 Effect of each new component.* Ablation studies on UniFreqSDG components are detailed in Tab. 4, using a plain ResNet-based segmentation model as the baseline on the Fundus dataset.

**Table 2: DSC Comparison of State-Of-The-Art (SOTA) methods on Prostate segmentation task [56]. We mark the top results in bold.**

| Methods | Prostate Segmentation (DSC ↑) | | | | | | Avg. DSC ↑ |
|---|---|---|---|---|---|---|---|
| | A to Rest | B to Rest | C to Rest | D to Rest | E to Rest | F to Rest | |
| ERM [50] | 63.73 | 61.21 | 27.41 | 34.36 | 44.10 | 61.70 | 48.75 |
| RandConv[66] (ICLR'21) | 75.52 | 57.23 | 44.21 | 61.27 | 49.98 | 54.21 | 57.07 |
| CSDG[42] (TMI'22) | 80.72 | 68.00 | 59.78 | 72.40 | 68.67 | 70.78 | 70.06 |
| MaxStyle[8] (MICCAI'22) | 81.25 | 70.27 | 62.09 | 58.18 | 70.04 | 67.77 | 68.27 |
| EFDM[71] (CVPR'22) | 80.87 | 69.78 | 63.16 | 65.39 | 69.84 | 67.15 | 69.37 |
| SLAug[51] (AAAI'23) | 81.20 | 69.32 | 60.92 | 73.72 | 67.15 | 71.93 | 70.71 |
| TriD[13] (MICCAI'23) | 81.50 | 70.28 | 62.89 | 74.52 | 72.12 | 69.11 | 71.74 |
| Aloft-E[23] (CVPR'23) | 81.61 | 72.05 | 64.26 | 79.23 | 73.04 | 72.25 | 73.74 |
| $UniFreqSDG_s$ | $81.79_{\pm0.23}$ +0.29 | $72.19_{\pm0.12}$ +1.91 | $64.58_{\pm0.13}$ +1.69 | $79.17_{\pm0.34}$ +4.65 | $73.18_{\pm0.39}$ +1.06 | $72.39_{\pm0.29}$ +3.28 | **73.88** +2.14 |
| $UniFreqSDG_m$ | $82.55_{\pm0.12}$ +1.05 | $77.17_{\pm0.07}$ +6.89 | $66.11_{\pm0.21}$ +3.22 | $82.35_{\pm0.18}$ +7.83 | $75.86_{\pm0.26}$ +3.74 | $73.74_{\pm0.38}$ +4.63 | **76.30** +4.56 |
| $UniFreqSDG_l$ | $82.79_{\pm0.14}$ +1.29 | $77.65_{\pm0.16}$ +7.37 | $67.18_{\pm0.56}$ +4.29 | $82.51_{\pm0.15}$ +7.99 | $76.12_{\pm0.32}$ +4.0 | $74.13_{\pm0.27}$ +5.02 | **76.73** +4.99 |

**Table 3: SDG classification on PACS [48]. We train the model on one of these source domains and evaluate the model on the rest domains.**

| Methods | Venue | P | A | C | S | Avg. ACC. |
|---|---|---|---|---|---|---|
| ERM [50] | - | 33.65 | 65.38 | 64.20 | 34.15 | 49.34 |
| ERM w/ MAD [49] | CVPR'23 | 32.32 | 66.47 | 69.80 | 34.54 | 50.78 |
| Augmix [25] | ICLR'19 | 38.30 | 66.54 | 70.16 | 52.48 | 56.87 |
| pAdaln [41] | CVPR'21 | 33.66 | 64.96 | 65.24 | 32.04 | 48.98 |
| Mixstyle [77] | ICLR'21 | 37.44 | 67.60 | 70.38 | 34.57 | 52.50 |
| DSU [34] | ICLR'22 | 42.10 | 71.54 | 74.51 | 47.75 | 58.97 |
| ACVC [16] | CVPR'22 | 48.05 | 73.68 | 77.39 | 55.30 | 63.61 |
| ACVC w/ MAD [49] | CVPR'23 | 52.95 | 75.51 | 77.25 | 57.75 | 65.87 |
| $UniFreqSDG_s$ | - | 55.02 +2.07 | 76.60 1.09 | 78.19 +0.94 | 63.35 +5.60 | 68.29 +2.42 |
| $UniFreqSDG_m$ | - | 55.65 2.70 | 77.43 1.92 | 80.42 +3.17 | 66.38 +8.63 | 69.97 +4.10 |
| $UniFreqSDG_l$ | - | 56.97 +4.02 | 78.94 +3.43 | 79.51 +2.26 | 68.92 11.17 | 70.59 +4.72 |

**Table 5: Performance comparisons of different inserted positions of our UniFreqSDG method on the Fundus segmentation tasks [13].**

| Encoder | | | | Decoder | | | | Fundus Dataset |
|---|---|---|---|---|---|---|---|---|
| $E_1$ | $E_2$ | $E_3$ | $E_4$ | $D_1$ | $D_2$ | $D_3$ | $D_4$ | Avg. DSC ↑ |
| ✓ | - | - | - | - | - | - | - | 83.98 |
| - | ✓ | - | - | - | - | - | - | 83.56 |
| - | - | ✓ | - | - | - | - | - | 83.43 |
| - | - | - | ✓ | - | - | - | - | 83.19 |
| ✓ | ✓ | - | - | - | - | - | - | 84.09 |
| ✓ | ✓ | ✓ | - | - | - | - | - | 84.60 |
| ✓ | ✓ | ✓ | ✓ | - | - | - | - | **85.31** |
| ✓ | ✓ | ✓ | ✓ | ✓ | - | - | - | 82.17 |
| ✓ | ✓ | ✓ | ✓ | - | ✓ | - | - | 83.66 |
| ✓ | ✓ | ✓ | ✓ | - | - | ✓ | - | 83.33 |
| ✓ | ✓ | ✓ | ✓ | - | - | - | ✓ | 82.96 |

**Table 4: Ablation experiments on each component in UniFreqSDG for the Fundus segmentation [13].**

| Variants | LSP | CPR | $\mathcal{L}_{dec}$ | $\mathcal{L}_{sty}$ | $\mathcal{L}_{con}$ | Fundus | Prostate |
|---|---|---|---|---|---|---|---|
| Baseline | | - | - | - | - | 80.32 | 60.51 |
| Variant1 | ✓ | - | - | - | - | 83.45 | 73.89 |
| Variant2 | ✓ | ✓ | - | - | - | 83.98 | 74.67 |
| Variant3 | ✓ | ✓ | ✓ | - | - | 84.23 | 75.68 |
| Variant4 | ✓ | ✓ | - | ✓ | - | 84.12 | 75.92 |
| Variant5 | ✓ | ✓ | - | - | ✓ | 84.27 | 75.82 |
| Variant6 | ✓ | ✓ | ✓ | ✓ | - | 84.45 | 75.98 |
| Variant7 | ✓ | ✓ | ✓ | - | ✓ | 84.35 | 76.12 |
| Variant8 | ✓ | ✓ | - | ✓ | ✓ | 84.54 | 76.02 |
| $UniFreqSDG_m$ | ✓ | ✓ | ✓ | ✓ | ✓ | **85.31** | **76.30** |

Results demonstrate that the addition of Learnable Spectral Perturbation (LSP) and Content Preservation Reconstruction (CPR) notably boosts performance. The inclusion of $\mathcal{L}_{dec}$, $\mathcal{L}_{sty}$, and $\mathcal{L}_{con}$, components of the Active Domain-variance Inducement Loss, each

positively impacts performance. Variants 3 through 8 compare the effects of these loss components. Omitting any of these losses decreases performance, underscoring that ADI loss effectively guides LSP to generate diverse style samples and helps the model adapt to style variations in OOD samples, thereby enhancing generalization.

*4.3.2 Effect of different perturbation injected positions.* To evaluate the impact of perturbation injection locations on SDG performance, perturbations were introduced at various positions within the hierarchical feature maps of UniFreqSDG. The findings, presented in Tab. 5, indicate that perturbation injection in the encoder consistently improves model performance over the baseline across all locations. Conversely, perturbations in the decoder negatively affect performance, likely due to its proximity to the output, where perturbations can more significantly influence the results.

*4.3.3 T-SNE visualization for UniFreqSDG.* In Fig. 6, we use t-SNE visualization [52] to compare the feature separability of UniFreqSDG with that of the TriD method [13]. The visualization shows that, in the first column, the category features differentiated by TriD have poorer discernibility, likely due to inadequate learning capabilities for handling OOD conditions. This results in an inability to adapt to significant style variations. In the second column,

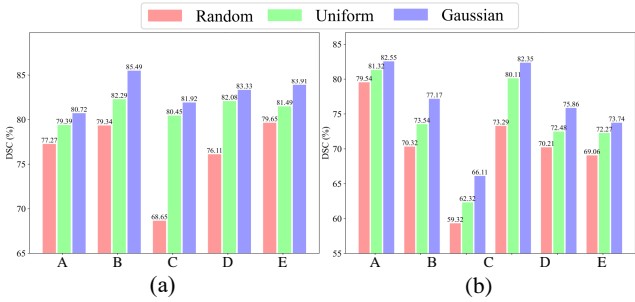

**Figure 5: Ablation study for different perturbation noise distributions on two datasets: (a) Fundus [13], (b) Prostate [56].**

**Table 6: Performance comparisons of different implementations of the decoupling module on Fundus dataset [13].**

| Decoupling | Optical Disc / Cup Segmentation (DSC ↑) | | | | | Avg. DSC ↑ |
|---|---|---|---|---|---|---|
| | A | B | C | D | E | |
| SA | 83.16 | 86.59 | **85.39** | 83.75 | 85.36 | 84.96 |
| CA | 80.97 | 87.11 | 83.54 | 84.38 | **85.50** | 84.30 |
| Self-attention | 82.17 | 85.40 | 82.49 | 83.99 | 84.26 | 83.66 |
| ECA | 81.54 | **87.14** | 84.69 | 84.77 | 84.39 | 84.51 |
| CA + SA | **83.82** | 86.27 | 84.84 | **85.36** | 85.31 | **85.31** |

features from different domains using the TriD method tend to cluster together, indicating a lack of discriminative ability across categories. Conversely, UniFreqSDG demonstrates effective categorization across different domains. Features from different domains and categories are well-separated within the distribution contours marked by *magenta* and *violet* dashed lines, illustrating UniFreqSDG's capability to distinguish data from unseen domains.

*4.3.4 Different distributions for perturbing.* In this work, we default to using noise sampled from a Gaussian distribution for perturbation injection. Here, we also consider other noise distributions: Random Gaussian Distribution (referred to as Random) and the Uniform Distribution (referred to as Uniform). For the Random distribution, we sample random noises from a Gaussian distribution $N(0, 1)$ and add them to the LF components. For the uniform distribution, noise is sampled from $U(-\Theta, +\Theta)$, with $\Theta$ being the variance defined in Eq. 8. As shown in Fig. 5, overall, noise injection using the Gaussian Distribution exhibits the best performance.

*4.3.5 The ablation study of different decoupling module.* In CPR module, the "decoupling-reconstruction" operation successfully extracts category-relevant, domain-invariant features from both channel and spatial dimensions, ensuring content preservation during LF perturbations. We evaluated several alternative modules, including Spatial Attention (SA) [59], Channel Attention (CA) [26], Efficient Channel Attention (ECA) [55], and the Self-attention mechanism [17]. As presented in Tab. 6, our approach, which facilitates interactions across channels and pixels, outperforms these alternative decoupling methods. Furthermore, the experimental findings suggest that the decoupling module does not significantly alter the performance of our method, underscoring the effectiveness and robustness of our overall framework.

*4.3.6 Feature-level visualization.* We also visualized the fundus image features decoupled by the model, including content and style

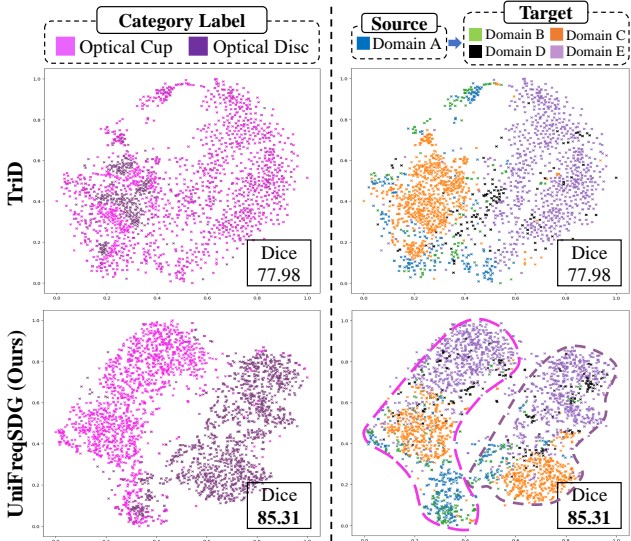

**Figure 6: The t-SNE visualization. Colors are used to indicate categories in the first column and domains in the second column. The first column demonstrates that our method possesses stronger recognition capabilities for segmentation structures, evident from the distinct color groupings representing different categories. The second column illustrates that our method can more effectively reduce the gap between different domains (the *magenta* and *violet* dashed lines represent the distribution contours of two categories).**

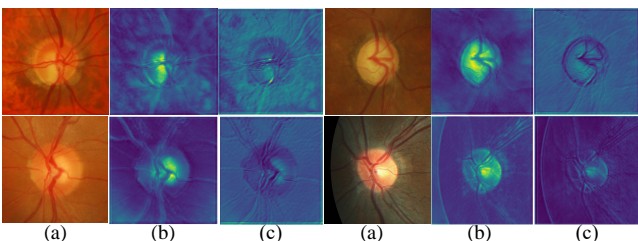

**Figure 7: Visualization of decoupled features in the CPR module: (a) input, (b) content feature, and (c) style feature.**

features, as shown in Fig. 7. From the figure, it is observable that the content feature focuses more on the fundus-related Optical cup and disc within the image, whereas the style feature pays more attention to variations in brightness and background information.

## 5 Conclusion

In this work, we propose the ***UniFreqSDG*** for single-source domain generalization in medical imaging. By innovatively designing Learnable Spectral Perturbations, UniFreqSDG adaptively introduces low-frequency perturbations into hierarchical features based on the input data. The Content Preservation Reconstruction strategy is designed to prevent the loss of category-related content information, and an Active Domain-Variance Inducement strategy is introduced to ensure the effectiveness and efficiency of the entire framework. Extensive experiments conducted across three datasets demonstrate the universality, flexibility, and generalization capabilities of our method.

## Acknowledgments

This work was supported in part by the National Key Research and Development Program of China under Grant No.2021ZD0140407, the Beijing Natural Science Foundation L222152 and the National Natural Science Foundation of China under Grant No. U21A20523.

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
