# OpenReview forum: "Universal Frequency Domain Perturbation for Single-Source Domain Generalization"
_acmmm.org/ACMMM/2024/Conference — MM2024 Poster_

### Official Review · Reviewer_ejxu · 2024-05-06

**Rating:** 4
**Confidence:** 2

**Summary:**

This paper introduces UniFreqSDG, a novel approach for single-source domain generalization (SDG) in medical imaging that addresses the challenge of style variation in out-of-distribution (OOD) domains. It performs adaptive frequency domain perturbations to generate diverse samples while preserving domain-invariant features. The method outperforms state-of-the-art approaches on fundus and prostate datasets.

**Strengths:**

1.Single domain generalization is a practical and essential topic in medical image segmentation, making this work relevant and important.

2.The proposed method is technically sound and well-designed.

3.The performance of the proposed method is impressive, demonstrating its effectiveness.

**Limitations:**

1.Different baselines use varying backbone architectures, such as ResNet18 and ResNet34. To facilitate a fair comparison of the performance across different methods, a column specifying the backbone used for each method should be included in the results table.

2.Since the main baseline, TriD, is based on ResNet34, it is necessary to conduct an ablation study using ResNet34 as the backbone for the proposed model. This will help verify the effectiveness of the proposed model's components and ensure that the performance improvements are not solely due to differences in the backbone architecture.

3.The related works section is missing some relevant papers on domain generalization in medical imaging, such as [1] and [2]. Including these papers and discussing how the proposed method compares to or builds upon their contributions would strengthen the paper's context within the existing literature.

[1] Prompt-driven latent domain generalization for medical image classification

[2] Contrastive Single Domain Generalization for Medical Image Segmentation

**Suitability:**

2

---

### Official Review · Reviewer_BTMs · 2024-05-09

**Rating:** 4
**Confidence:** 4

**Summary:**

This paper improves the domain generalization ability of models by adding perturbations into frequency decomposed features. The low-frequency regions are predicted by learnable layers and are different across channels. The experiment results demonstrate the efficacy of the proposed method.

**Strengths:**

1. The style information are decoupled twice. The perturbation is only injected into the low-frequency part and a content-style decoupling approach is also used.
2. The proposed methods can be easily adapted to other medical image segmentation tasks.
3. The improvement is significant compared with existing methods.

**Limitations:**

1. Only DSC is used as evaluation metric, while in other domain generalization papers, HSD is also widely used.
2. The experiment includes three CNN backbones, while it is better to compare with more architectures such as transformers or recent foundation models.
3. It appears that the proposed method can be applied to other domain generalization settings. For instance, does the effectiveness of the method decrease as the number of source domain datasets increases?

**Suitability:**

3

---

### Official Review · Reviewer_dicx · 2024-05-26

**Rating:** 3
**Confidence:** 3

**Summary:**

The paper proposes the UniFreqSDG for single-source domain generalization in medical imaging. UniFreqSDG introduces low-frequency perturbations into intermediate features and uses Content Preservation Reconstruction to preserve content information. Additionally, it employs an Active Domain-Variance Inducement to ensure the effectiveness and efficiency of the entire framework. Extensive experiments conducted across three datasets showcase the capabilities of UniFreqSDG.

**Strengths:**

The paper is well-structured, providing comprehensive explanations and validations for each module and detail. Extensive experiments are conducted on two medical datasets and one natural image dataset, validating the effectiveness of its method.

**Limitations:**

The paper shares a similar approach with ALOFT [1] in its LSP module, both extracting the low-frequency component in the frequency domain and subjecting it to Gaussian statistical perturbations. It might be worth referencing or comparing with ALOFT [1].

1) Why is there not much improvement in performance on both two medical benchmarks when using different scales of M/L, given that ResNet-34/50 is used as the backbone?

2) There is a lack of comparison with more single-source domain methods in 2023, such as CCSDG [2], which has shown excellent performance on the fundus dataset.

3) How dose the loss L_dec work? Why use the Gram matrix in L_dec and L_sty?

4) The paper contains several citation errors and formatting issues, including incorrect table references like "As presented in Table 8" in line 893 and "As shown in Fig. 7" in line 1303 of the supplementary material. Additionally, there are improperly formatted citations, such as [52].

Reference:
[1] ALOFT: A Lightweight MLP-like Architecture with Dynamic Low-frequency Transform for Domain Generalization
[2] CCSDG: Devil is in Channels: Contrastive Single Domain Generalization for Medical Image Segmentation

**Suitability:**

2

---

### Meta-Review · Area_Chair_7hJF · 2024-07-05

**Recommendation:** Accept (Poster)
**Confidence:** 4

**Metareview:**

The study introduces the UniFreqSDG framework, a novel approach to enhancing single-source domain generalization (SDG) in medical imaging through hierarchical frequency domain perturbations. Reviewers appreciated the well-structured experimentation and improved performance demonstrated on medical datasets. However, they noted several limitations, including a lack of comparisons with recent methods, a need for clearer ablation studies, and clarification issues. The authors satisfactorily addressed most concerns in their rebuttal, clarifying methodological details and providing additional experimental results, leading to a consensus among reviewers to recommend acceptance. Thus, the final decision is to accept the paper. The authors are advised to carefully revise the paper based on the reviewers' comments and rebuttal content.